# Prevalence and predictors of musculoskeletal injuries among gym members in Bangladesh: A nationwide cross-sectional study

Mohammad Jahirul Islam[1], Md. Selim Rana[2], Md. Sharifuddin Sarker[3], Md. Mahemanul Islam[4], Md. Nuruzzaman Miah[5], Md. Anwar Hossain[6], Ruwaida Jahangir[7], Rahemun Akter[8], Sohel Ahmed[8,9]*

1 Department of Physical Medicine and Rehabilitation, MAG Osmani Medical College Hospital, Sylhet, Bangladesh, 2 Department of Physiotherapy, Chittagong Institute of Medical Technology, Agrabad, Chittagong, Bangladesh, 3 Department of Physiotherapy, Sajida Hospital, Sajida Foundation, Dhaka, Bangladesh, 4 Department of Physiotherapy, Progetto Uomo Rishilpi International Onlus, Khulna, Bangladesh, 5 The Popular Consultation and Physiotherapy Center, Narayanganj, Dhaka, Bangladesh, 6 The Barekha Hospital Madonpur Limited, Narayanganj, Dhaka, Bangladesh, 7 Department of Physiotherapy, Gonoshasthay Nagar Hospital, Dhaka, Bangladesh, 8 Ahmed Physiotherapy & Research Center, Dhaka, Bangladesh, 9 Directorate of Student's Welfare, Bangladesh University of Engineering and Technology, Dhaka, Bangladesh

☯ These authors contributed equally to this work.
* ptsohel@gmail.com

**Data Availability Statement:** All relevant data are within the paper and its Supporting information files.

## Abstract

### Background

Participating in physical exercise is advantageous for maintaining optimum health, improving physical capacity, decreasing the likelihood of chronic diseases, and promoting overall wellbeing.

### Aim

This study aimed to find out the prevalence and factors that contribute to musculoskeletal injuries among individuals who participated in fitness activities at the gym.

### Methods

This cross-sectional study included 1123 gym members, both male and female, aged between 18 and 50 years, from selected fitness centers in Bangladesh. Musculoskeletal injuries were assessed using the Nordic musculoskeletal disorder questionnaire. Binary logistic regression identified the gym members' predictors of musculoskeletal injuries.

### Results

The highest prevalence of musculoskeletal injuries at the low back (36.6%) was seen among the eight body sites, followed by the shoulder (24.7%) and knee (17.1%). Males (aOR 2.589, CI 1.18 to 5.65) and those who go to the gym to lose weight (aOR 3.859, CI 0.91 to 16.33) and for physical fitness (aOR 1.895, CI 1.07 to 3.35) had a greater risk of musculoskeletal injury. Participants who carried out strength training exercises (aOR 4.10, CI

**Funding:** The author(s) received no specific funding for this work.

**Competing interests:** The authors have clarified that there are no competing interests.

2.74 to 6.19) had a four-fold increased risk of musculoskeletal injuries than those who did not. Furthermore, higher adjusted odds of musculoskeletal injuries were found for the potential causes of injuries in incorrect holding (aOR 1.69, CI 1.10 to 2.60), overweight lifting (aOR 2.00, CI 1.30 to 3.08), lack of workout knowledge (aOR 3.56, CI 2.09 to 5.85), and insufficient information from the trainer (aOR 5.66, CI 1.84 to 17.39).

## Conclusion

Musculoskeletal injuries are highly prevalent among gym-goers in Bangladesh. The back was the most often injured area, followed by the shoulder and knee. It is important to exhibit caution and take extra care while doing strength training activities in order to avoid injury. Prior to engaging in gym-based activities, it is essential to have a thorough understanding of proper exercise knowledge.

## Introduction

Engaging in physical exercise is beneficial for maintaining optimal health, enhancing physical performance, reducing the risk of chronic diseases, and enhancing overall well-being [1, 2]. Fitness centers serve as a vital hub for engaging in physical activity, where members tend to exercise more often than they typically do in their own homes [3, 4]. In 2016, the gym-based exercise participation rate among Dutch residents aged 12–79 was 21 percent, which is an increase from 12 percent in 2001 [5]. Gym-based fitness activities provide a range of exercise facilities in a secure environment, promoting social engagement, and giving members access to expert's exercise instruction and coaching [3]. While engaging in physical activity via sports is generally seen as advantageous, participating in fitness activities in the gym is linked to the potential for injuries [6]. A descriptive epidemiological study in the Netherlands shows that many injuries occur as a result of gym-based fitness activities [6]. In 2019, the total number of reported injuries resulting from fitness activities in gyms amounted to 860,000, which constituted 16% of all sports injuries in the Netherlands [7]. A descriptive epidemiological study in Maylasia reported higher injury rates in athletes from nonaffiliated training gyms compared to CrossFit-affiliated gyms [8].

A study conducted in the Netherlands revealed that most injuries occurring during gym workouts were a result of strength training. The shoulder, leg, and knee were the main anatomical regions injured, with 73.1% of the reported injuries attributed to unsupervised gym-based exercise activities [6]. Another study in South Florida revealed that out of 191 athletes, 50 individuals had a collective total of 62 injuries while participating in CrossFit over a period of 6 months [9]. Research conducted in Saudi Arabia examined the occurrence of injuries among individuals who are members of gyms. The survey revealed that 29.2% of the participants reported experiencing musculoskeletal injuries. The shoulder accounted for 40.5% of injuries, followed by the foot at 32.4%, and the back at 25.7% [10].

The majority of patients treated in the emergency department among gym-goers were due to exercise-related injuries [10], as well as injuries associated with particular exercise programs such as CrossFit [11–13], powerlifting, and weightlifting [14, 15]. Various fitness activities, such as CrossFit and power- and weightlifting, have undergone thorough investigation in several countries. The recent systematic reviews have collated and examined the results of these investigations [12–14]. In a study conducted by Lubetzky-Vilnai A et.al., it was shown that

41.6% of the total participants (190 out of 457 persons) reported experiencing exercise-related injuries in the gym [16]. The published results are limited to certain age groups, specific fitness activities performed in gyms, and injuries associated with fitness that were treated in emergency rooms. Nevertheless, these rules do not include the whole population involved in fitness activities or those specifically conducted in gyms. In order to provide appropriate guidance and education on preventing gym-related injuries, it is essential to collect extensive data on these injuries across the wider recreational exercise community. Specific data about the total number of injuries resulting from workout activities in gyms can be obtained from countries such as the Netherlands [6] and Saudi Arabia [10]. Nevertheless, the comprehension of this information is restricted, and there is a paucity of data regarding the prevalence and predictors of gym-related injuries. There is a lack of data on the prevalence of musculoskeletal injuries among gym members and the predictive risk factors of those injuries in Bangladesh. This study is the first to tackle this significant issue in Bangladesh. The objective of this research was to determine the prevalence and factors that contribute to musculoskeletal injuries among individuals who participated in fitness activities at gyms in Bangladesh.

## Materials and methods

### Study design and participants

A descriptive cross-sectional study was conducted in Bangladesh from June 25 to December 30, 2022, to collect information about individuals who participate in fitness activities in gyms. The data was collected via face-to-face interviews. Due to the lack of available data on the total number of gym centers in Bangladesh and the number of individuals who regularly attend gyms, we conducted a random visit to gym facilities in five cities. In order to streamline data collection, we made a single visit to each gym facility and gathered data from the fitness trainees who were there at the time.

### Ethical statement

This study adhered to the ethical standards for medical research with human participants as outlined in the World Medical Association Declaration of Helsinki (Revised 2013). The ethical review board of Sylhet MAG Osmani Medical College Hospital (Approval No: ICE/physio/22/008) approved the study protocol. The study rigorously followed the Strengthening the Reporting of Observational Studies in Epidemiology guideline for cross-sectional studies [S1 Checklist]. Before commencing the individual interview session, written informed consent was obtained by offering a comprehensive explanation of the study's aim, objectives, potential benefits, and dangers. The participants were also informed of the voluntary nature of the interview and their participation.

### Subject selection criteria

The research included participants between the ages of 18 and 50, of both genders, who went to the gym for at least six months and who agreed to participate. The research eliminated those with a history of non-gym-related injuries, as well as those with musculoskeletal or neurological diseases, and participants who were using steroids.

### Sample and sample size

The necessary sample size for this cross-sectional study was determined using the method for determining proportions: $n = Z\alpha^2 P (1 - P)/d^2$; where $Z\alpha = 1.96$; $P$ = the expected musculoskeletal injuries reported in the previous literature; as we don't have any previous literature

available regarding this issue, we assume a 50% response distribution and $d$ = 3% marginal error [17]. Thus, the study found that a minimum sample size of 1067 was required. Based on an anticipated 10% incidence of incomplete forms, we have calculated that the absolute minimum required sample size should be 1173. A total of 1250 people were interviewed, resulting in the attainment of the final sample. Nevertheless, 44 individuals failed to respond to all the questions, so we excluded them from the final analysis.

## Sampling

Data were collected from all five geographic divisions of Bangladesh, and a dual-stage cluster sampling technique was used to include potential samples. We randomly chose the gym center as clusters in the first stage. Due to the lack of available data on the total number of gym centers in Bangladesh and the number of individuals who regularly attend gyms, we conducted a random visit to 74 fitness centers spread over five administrative divisions, including metropolitan and semi-urban areas. In order to streamline data collection, we made a single visit to each gym facility and gathered data from the fitness trainees who were there at the time.

## Survey questionnaire

The questionnaire encompasses demographic data (age, gender, height, weight, BMI, and occupation), workout-related details (reason for joining the gym, daily workout duration, types of exercises performed at the gym, adequacy of the instructor's attention during workouts, and inclusion of warm-up/cool-down activities), and injury-related information (history of previous workout-related injuries, possible causes of injury, current pain status, continuation of workouts after an injury, adherence to rest periods following an injury, and injury management strategies) [S1 Text].

The Nordic musculoskeletal disorder questionnaire served as a tool for pinpointing the location of pain. The questionnaire included eight distinct anatomical locations where pain was experienced, namely the neck, shoulder, elbow, wrist/hand, back, hip/thigh, knee, and ankle/foot. The Nordic musculoskeletal questionnaire's validity has been previously demonstrated in existing research [18]. The questionnaire's sensitivity for detecting pain in the body ranged from 66% to 92%, while its specificity ranged from 71% to 88% [19].

## Data collection procedure

The survey was conducted using an interviewer-administered questionnaire. Eight data collectors, all of whom have graduated in physiotherapy, were selected for the purpose of data collection. The interview was conducted in Bangla. Data was gathered from five administrative divisions (Dhaka, Sylhet, Rajshahi, Khulna, and Chittagong) of Bangladesh, where 74 fitness facilities of different divisional cities, districts and thana-level gym centers were visited for the purpose of data collection. Illegible participants were requested to participate in this study. Informed consent to collect, analyze, and publish their data anonymously was obtained before starting an interview. Upon receiving authorization, a data collector conducted an interview and completed a questionnaire using a paper-based format. Initially, the data collector posed a question to the respondent, and upon receiving an answer, the data collector repeated the response to the respondent in order to get confirmation.

## Statistical analysis

The descriptive data were presented as frequency, percentage, mean, and standard deviation, where applicable. In order to establish the correlation between musculoskeletal pain and

sociodemographic characteristics, the Pearson's Chi-square test or Fisher's exact test were used. Multiple logistic regression analysis was conducted to calculate adjusted and unadjusted odds ratios with a 95% confidence interval. The independent variables in this study were the presence of pain, socio-demographic characteristics, and workout-related information as predictor variables for musculoskeletal pain. The variables were found to be statistically significant in the descriptive analysis were included in the regression model for the computation of adjusted odds ratios (aORs). The fitness of the model was checked by Hosmer-Lemshow's test and classification table. The presence of multicollinearity among the independent variables was assessed using variance inflation factors (VIFs), with a predetermined threshold of VIF$\leq$5.0. The significance level was established at <0.05. The statistical analysis was conducted using IBM SPSS Version 25.

## Results

### Participants' general characteristics

In total, this study examined the data of 1123 individuals. The average age, weight, height, and Body Mass Index (BMI) of the participants were 27.16 ± 6.58 years, 69.60 ± 10.67 kg, 168.59 ± 7.26 cm, and 24.28 ± 3.58 kg/m2, respectively. Table 1 presents a statistical summary of the sociodemographic characteristics. Out of the total respondents, 1039 (92.0%) were male, and 538 (47.7%) belonged to the age range of 18–25 years, and 458 (40.6%) were service holder. Based on the BMI Asian classification, 442 (39.1%) of the individuals in the study were classified as obese, while 615 (54.5%) of the participants reported going to the gym to maintain their physical fitness. Out of the total participants, about 540 (47.8%) worked out for at least an hour in the gym. Additionally, 88.2% of the participants incorporated warm-up and cool-down exercises into their routine. Of them, 67.8% did aerobic activities, 67.5% strength training, 50.0% powerlifting, and 42.8% CrossFit exercises. 25% of participants ascribed their injuries to excessive exercise, 19.9% to improper grip, 22.1% to lifting weights above their capacity, 15.3% to insufficient knowledge of exercise techniques, 5.6% to inadequate guidance from the trainer, and 16.3% to a lack of caution during exercise.

### Prevalence of musculoskeletal injuries

The highest prevalence of musculoskeletal injuries at the low back (17.2%) was seen among the eight body sites, followed by the shoulder (11.0%) and knee (7.6%). Moreover, 3.8% of the participants reported neck injuries, whereas 6.5% and 4.6% reported hip/thigh and ankle/foot injuries, respectively. "Fig 1"shows the prevalence of musculoskeletal pain in various body regions among the participants. 34.5% of participants had encountered an injury at least once, whereas 25.6% of participants had experienced many instances of injuries during previous workout sessions. 49.1% of participants continued their workout after their injury, and 63.6% of the participants left their injury untreated or managed by themselves.

### Result of regression analysis

Males (aOR 2.589, CI 1.18 to 5.65) and those over 45 years of age (aOR 1.069, CI 0.19 to 6.03) had higher chances of musculoskeletal injury. Furthermore, those who go to the gym to lose weight (aOR 3.859, CI 0.91 to 16.33) and for physical fitness (aOR 1.895, CI 1.07 to 3.35) had a greater risk of musculoskeletal injury. Participants who carried out strength training exercises (aOR 4.10, CI 2.74 to 6.19) had a four-fold increased risk of musculoskeletal injuries than those who did not. Also, people who held their weight incorrectly (aOR 1.69, CI 1.10 to 2.60), lifted too much (aOR 2.00, CI 1.30 to 3.08), didn't know enough about the workout (aOR 3.56,

**Table 1. Socio-demographic characteristics of the participants.**

| Variable | | Total sample (n, %) | No pain (n, %) | Presence of pain (n, %) | P-value |
|---|---|---|---|---|---|
| Total sample | | 1129 (100) | 627 (55.5) | 502 (44.5) | |
| **Gender** | | | | | 0.008 |
| Male | | 1039 (92.0) | 565 (50.0) | 474 (42.0) | |
| Female | | 90 (8.0) | 62 (5.5) | 28 (2.5) | |
| **Age group** | | | | | <0.001 |
| 18–25 | | 538 (47.7) | 348 (30.8) | 190 (16.8) | |
| 26–35 | | 493 (43.7) | 246 (21.8) | 247 (21.9) | |
| 36–45 | | 86 (7.6) | 30 (2.7) | 56 (5.0) | |
| >45 | | 12 (1.1) | 3 (0.3) | 9 (0.8) | |
| **BMI (Asian classification)** | | | | | <0.001 |
| Underweight (<18.5) | | 23 (2.0) | 16 (1.4) | 7 (0.6) | |
| Normal weight | | 356 (31.5) | 228 (20.8) | 128 (11.3) | |
| Overweight | | 308 (27.3) | 167 (14.8) | 141 (12.5) | |
| Obese | | 442 (39.1) | 216 (19.1) | 226 (20.0) | |
| **Occupation** | | | | | <0.001 |
| Student | | 432 (38.3) | 290 (25.7) | 142 (12.6) | |
| Business | | 169 (15.0) | 86 (7.6) | 83 (7.4) | |
| Service | | 458 (40.6) | 222 (19.7) | 236 (20.9) | |
| Homemaker | | 23 (2.0) | 9 (0.8) | 14 (1.2) | |
| Unemployed | | 47 (4.2) | 20 (1.8) | 27 (2.4) | |
| **Purpose of GYM joining** | | | | | <0.001 |
| Losing weight | | 258 (22.9) | 97 (8.6) | 161 (14.3) | |
| Physical fitness | | 615 (54.5) | 363 (32.2) | 252 (22.3) | |
| Bodybuilding | | 244 (21.6) | 163 (14.4) | 81 (7.2) | |
| Recreation | | 12 (1.1) | 4 (0.4) | 8 (0.7) | |
| **Hours spent working out in a day** | | | | | 0.132 |
| < 1 hour | | 59 (5.2) | 25 (2.2) | 34 (3.0) | |
| 1 hour | | 540 (47.8) | 298 (26.4) | 242 (21.4) | |
| 2 hours | | 487 (43.1) | 282 (25) | 205 (18.2) | |
| 3 hours or more | | 43 (3.8) | 22 (1.9) | 21 (1.9) | |
| **Type of exercise performed at the gym** | | | | | |
| Cardio exercises | No | 363 (32.2) | 212 (18.8) | 151 (13.4) | 0.200 |
| | Yes | 766 (67.8) | 415 (36.8) | 351 (31.1) | |
| Strength training | No | 367 (32.5) | 148 (13.1) | 219 (19.4) | <0.001 |
| | Yes | 762 (67.5) | 479 (42.4) | 283 (25.1) | |
| Powerlifting | No | 565 (50.0) | 292 (25.9) | 273 (24.2) | 0.010 |
| | Yes | 564 (50.0) | 335 (29.7) | 229 (20.3) | |
| CrossFit | No | 646 (57.2) | 342 (30.3) | 303 (26.9) | 0.042 |
| | Yes | 483 (42.8) | 285 (25.2) | 198 (177.5) | |
| **Possible causes of injury** | | | | | |
| Over exercise | No | 847 (75.0) | 507 (44.9) | 340 (30.1) | <0.001 |
| | Yes | 282 (25.0) | 120 (10.6) | 162 (14.3) | |
| Wrong holding | No | 904 (80.1) | 541 (47.9) | 363 (32.2) | <0.001 |
| | Yes | 225 (19.9) | 86 (7.6) | 139 (12.3) | |
| Overweight lifting | No | 879 (77.9) | 540 (47.8) | 339 (30.0) | <0.001 |
| | Yes | 250 (22.1) | 87 (7.7) | 163 (14.4) | |

(*Continued*)

**Table 1.** (Continued)

| Variable | | Total sample (n, %) | No pain (n, %) | Presence of pain (n, %) | P-value |
|---|---|---|---|---|---|
| Lack of workout knowledge | No | 956 (84.7) | 594 (52.6) | 362 (33.1) | <0.001 |
| | Yes | 173 (15.3) | 33 (2.9) | 140 (12.4) | |
| Inadequate information from the trainer | No | 1066 (94.4) | 623 (55.2) | 443 (39.2) | <0.001 |
| | Yes | 63 (5.6) | 4 (0.4) | 59 (5.2) | |
| Fatigueness | No | 994 (88.0) | 592 (52.4) | 402 (35.6) | <0.001 |
| | Yes | 135 (12.0) | 35 (3.1) | 100 (8.9) | |
| Not cautious during exercise | No | 945 (83.7) | 584 (51.7) | 361 (32.0) | <0.001 |
| | Yes | 184 (16.3) | 43 (3.8) | 141 (12.5) | |
| **Do you perform worm up and cool down prior to exercise?** | | | | | <0.001 |
| Yes | | 996 (88.2) | 584 (51.7) | 412 (36.5) | |
| No | | 133 (11.8) | 43 (3.8) | 90 (8.0) | |
| **Previous history of injury during a workout** | | | | | <0.001 |
| Not at all | | 451 (39.9) | 336 (29.8) | 115 (10.2) | |
| Once | | 389 (34.5) | 228 (20.2) | 161 (14.3) | |
| Multiple time | | 289 (25.6) | 63 (5.6) | 226 (20.0) | |
| **Continue workout after any injury** | | | | | <0.001 |
| Yes | | 554 (49.1) | 274 (24.3) | 280 (24.8) | |
| No | | 575 (50.9) | 353 (31.3) | 222 (19.7) | |
| **How injury managed** | | | | | <0.001 |
| Leave it untreated | | 345 (30.6) | 247 (21.9) | 98 (8.7) | |
| Self-management | | 373 (33.0) | 239 (21.2) | 134 (11.9) | |
| Consult with physician | | 152 (13.5) | 56 (5.0) | 96 (8.5) | |
| Consult with physiotherapist | | 259 (22.9) | 85 (7.5) | 174 (15.4) | |

CI 2.09 to 5.85), didn't get enough information from the trainer (aOR 5.66, CI 1.84 to 17.39), and weren't careful during exercise (aOR 2.878, CI 1.77 to 4.66) had higher odds of musculoskeletal injuries. However, those who did not do a warm-up and cool-down before exercising had a substantially greater risk of musculoskeletal pain (aOR 2.671, CI 1.61 to 4.41). Table 2 gives the entire set of findings.

## Discussion

Musculoskeletal injuries are highly prevalent among gym members in Bangladesh. Out of the eight body locations, the low back had the highest occurrence of musculoskeletal injuries, followed by the shoulder and knee. Individuals who work out at the gym in an attempt to lose weight are more susceptible to musculoskeletal injuries reported in this present study. Similar to our findings, a study reported that division 1 college wrestlers who had rapid weight loss had a greater likelihood of sustaining in-competition injuries. Each percentage of body weight dropped by wrestlers resulted in an 11% higher risk of injury [20]. A systematic review study regarding exercise selection and common injuries in fitness centers reported the primary causes of musculoskeletal discomfort and injury risk are mostly attributed to overuse, inadequate recovery time between sessions, incorrect technique, insufficient training in these specific body parts, and the frequent use of heavy loads. Proper exercise selection in resistance training programs necessitates expert supervision and adherence to correct lifting techniques and training habits that take into account the anatomical and biomechanical patterns of the musculoskeletal structures [21]. The risk of musculoskeletal injuries was four times higher for

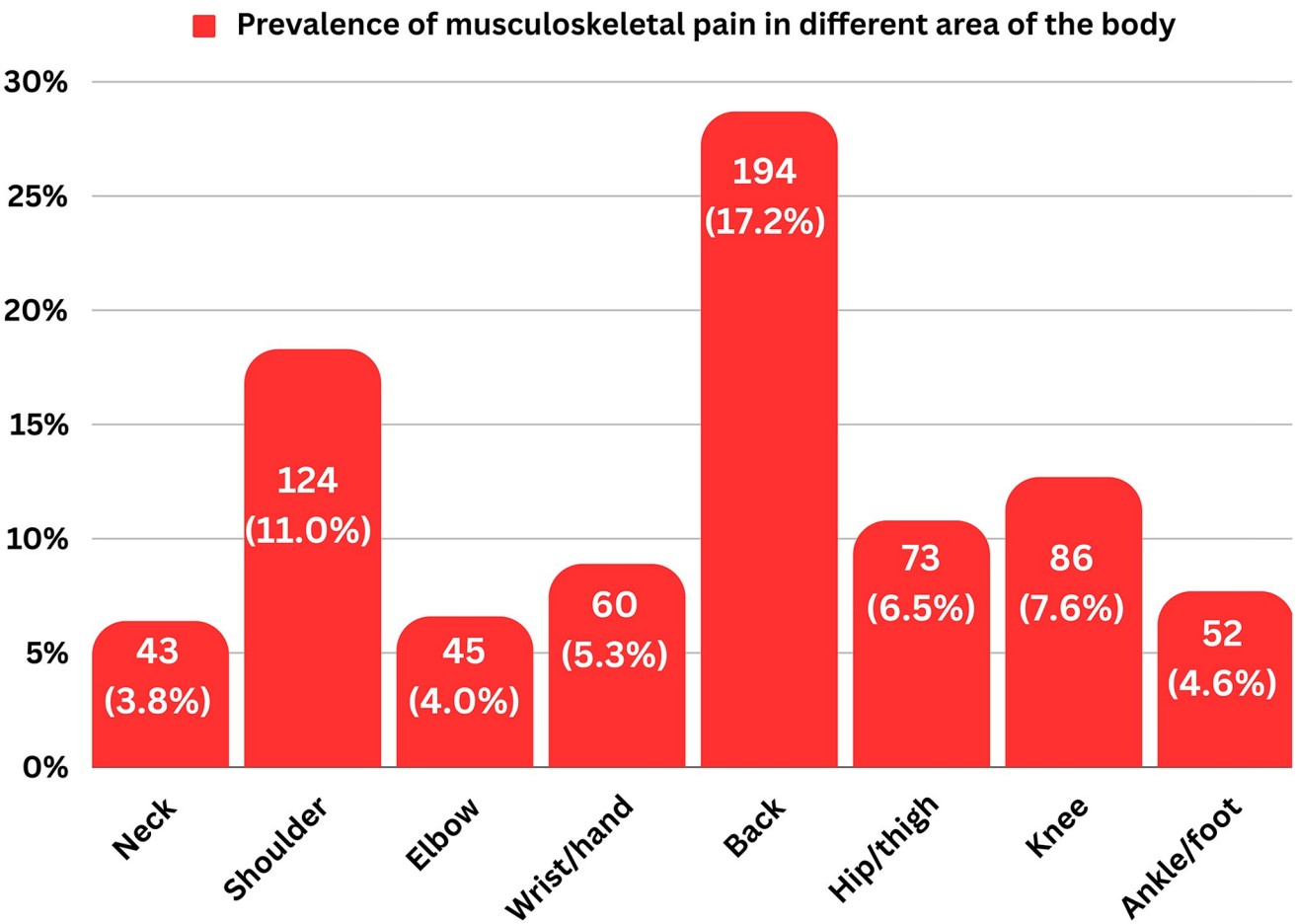

**Fig 1. Prevalence of musculoskeletal pain in various body regions.**

those who performed strength training activities compared to those who did not. In addition, there were increased possibilities of musculoskeletal injuries when it came to inappropriate gripping, lifting weights that were too heavy, not knowing how to work-out, and not receiving adequate advice from their trainer.

The most common site of injury was the back (17.2%), followed by the shoulder (11.0%) and the knee (7.6%). Similar to our result, a study conducted by Ahmed et al. reported that the back was the most commonly affected site for gym injuries, followed by the shoulder and knee [22]. In contrast to our study, Shinde and Sahasrabuddhe reported the shoulder as the most commonly affected site for gym injuries, followed by the lower back and knee [23]. The study by Feito et al. also reported that the shoulder was the most commonly affected site, followed by the back and knee [24].

Furthermore, this investigation revealed that males and those aged 45 and above had elevated susceptibility to musculoskeletal injury. Feito et al. observed a significantly higher incidence of injuries among men compared to women in their study [24]. Another study found a significant difference in gym-related injuries between men and women, with men exhibiting a much higher incidence of injuries in contrast to females. However, age was not proven to be a major contributing factor [10]. Nevertheless, several research studies have shown a lack of substantial disparity between males and females in terms of injury rates [8, 25, 26].

**Table 2. Multiple logistic regression analysis: Predictor of musculoskeletal injuries among the study participants.**

| Variable | | Unadjusted odd ratio | 95% CI | P-value | Adjusted odd ratio | 95% CI | P-value |
|---|---|---|---|---|---|---|---|
| **Gender** | | | | | | | |
| Female | | Reference | | | | | **0.017** |
| Male | | 1.858 | 1.17–2.95 | 0.009 | 2.589 | 1.18–5.65 | |
| **Age group** | | | | | | | |
| 18–25 | | Reference | | | | | |
| 26–35 | | 0.182 | 0.05–0.68 | 0.011 | 0.463 | 0.09–2.49 | 0.370 |
| 36–45 | | 0.335 | 0.09–1.25 | 0.104 | 0.593 | 0.11–3.12 | 0.537 |
| >45 | | 0.622 | 0.15–2.47 | 0.500 | 1.069 | 0.19–6.03 | 0.939 |
| **BMI** | | | | | | | |
| Underweight | | Reference | | | | | |
| Normal weight | | 0.418 | 0.17–1.03 | 0.060 | 0.780 | 0.25–2.46 | 0.671 |
| Overweight | | 0.537 | 0.40–0.71 | <0.001 | 0.747 | 0.48–1.14 | 0.180 |
| Obese | | 0.807 | 0.60–1.08 | 0.149 | 0.832 | 0.55–1.25 | 0.375 |
| **Occupation** | | | | | | | |
| Student | | Reference | | | | | |
| Business | | 0.363 | 0.19–0.67 | 0.001 | 0.349 | 0.15–0.81 | 0.014 |
| Service | | 0.715 | 0.37–1.37 | 0.313 | 0.491 | 0.20–1.18 | 0.113 |
| Homemaker | | 0.787 | 0.49–1.44 | 0.440 | 0.379 | 0.16–0.88 | 0.024 |
| Unemployed | | 1.152 | 0.41–3.18 | 0.785 | 0.754 | 0.19–2.94 | 0.684 |
| **Purpose of GYM joining** | | | | | | | |
| Recreation | | Reference | | | | | |
| Losing weight | | 4.025 | 1.17–13.76 | 0.026 | 3.859 | 0.91–16.33 | 0.067 |
| Physical fitness | | 3.340 | 2.31–4.82 | <0.001 | 1.895 | 1.07–3.35 | 0.028 |
| Bodybuilding | | 1.397 | 1.02–1.90 | 0.035 | 1.010 | 0.63–1.60 | 0.966 |
| **Type of exercise performed at the gym** | | | | | | | |
| Cardio exercises | Yes | 0.842 | 0.65–1.08 | 0.182 | 0.798 | 0.53–1.19 | 0.269 |
| | No | Reference | | | | | |
| Strength training | Yes | 2.505 | 1.94–3.23 | <0.001 | 4.10 | 2.74–6.19 | **<0.001** |
| | No | Reference | | | | | |
| Powerlifting | Yes | 1.368 | 1.08–1.73 | 0.009 | 1.38 | 0.92–2.08 | 0.119 |
| | No | Reference | | | | | |
| CrossFit | Yes | 1.279 | 1.01–1.62 | 0.043 | 1.13 | 0.78–1.66 | 0.498 |
| | No | Reference | | | | | |
| **Possible causes of injury while workout** | | | | | | | |
| Over exercise | Yes | 2.013 | 1.53–2.64 | <0.001 | 1.28 | 0.86–1.92 | 0.216 |
| | No | Reference | | | | | |
| Wrong holding | Yes | 2.409 | 1.78–3.25 | <0.001 | 1.69 | 1.10–2.60 | **0.017** |
| | No | Reference | | | | | |
| Overweight lifting | Yes | 2.984 | 2.22–4.00 | <0.001 | 2.00 | 1.30–3.08 | **0.002** |
| | No | Reference | | | | | |
| Lack of workout knowledge | Yes | 6.961 | 4.66–10.39 | <0.001 | 3.56 | 2.09–5.85 | **<0.001** |
| | No | Reference | | | | | |
| Inadequate information from the trainer | Yes | 20.74 | 7.48–57.52 | <0.001 | 5.66 | 1.84–17.39 | **0.002** |
| | No | Reference | | | | | |
| Fatigueness | Yes | 4.208 | 2.80–6.31 | <0.001 | 1.832 | 1.06–3.14 | **0.028** |
| | No | Reference | | | | | |

*(Continued)*

**Table 2.** (Continued)

| Variable | | Unadjusted odd ratio | 95% CI | P-value | Adjusted odd ratio | 95% CI | P-value |
|---|---|---|---|---|---|---|---|
| Not cautious during exercise | Yes | 5.305 | 3.68–7.64 | <0.001 | 2.878 | 1.77–4.66 | **<0.001** |
| | No | Reference | | | | | |
| **Perform worm up and cool down prior to exercise?** | | | | | | | |
| No | | 2.967 | 2.02–4.35 | <0.001 | 2.671 | 1.61–4.41 | **<0.001** |
| Yes | | Reference | | | | | |
| **Continue workout after any injury** | | | | | | | |
| Yes | | 1.625 | 1.28–2.06 | <0.001 | 1.127 | 0.81–1.57 | 0.476 |
| No | | Reference | | | | | |
| **How injury managed** | | | | | | | |
| Leave it untreated | | Reference | | | | | |
| Self-management | | 0.194 | 0.14–0.27 | <0.001 | 0.184 | 0.11–0.29 | <0.001 |
| Consult with physician | | 0.274 | 0.19–0.38 | <0.001 | 0.235 | 0.15–0.36 | <0.001 |
| Consult with physiotherapist | | 0.837 | 0.55–1.27 | 0.407 | 0.732 | 0.43–1.23 | 0.241 |

Our research found that those who participated in strength training activities had a fourfold increased probability of experiencing musculoskeletal injuries. These findings align with the data reported by Kemler et al. indicating that 52% of the participants engaged in strength training, and most of the injuries occurred during strength training activities [6]. According to Gray and Finch, 55% of the injuries occurring in fitness centers and requiring treatment in emergency rooms are specifically associated with strength training. Gray and Finch also examined the precise etiology of the injuries sustained [26]. They found overexertion or vigorous and/or abnormal movement emerged as a significant factor contributing to injuries in their research. Kemler et al. found that injury was most often attributed to overuse, overload, missteps and sprains, or poor posture or movement [6]. Our study also found a few contributing factors, for instance, overexercise, wrong holding, overweight lifting, a lack of workout knowledge, inadequate information from the trainer, and not being cautious during exercise. Gray and Finch conducted an analysis of injuries treated in emergency departments, whereas we examined all injuries experienced by a representative sample of individuals engaged in fitness activities. This disparity in methodology might account for the variation in identified causes.

In this study, individuals who had inadequate knowledge about exercises saw a four-fold increase in injuries, while those who did not get sufficient guidance from their trainer experienced a five-fold increase in injuries. A recent review has identified several factors that can contribute to injuries during resistance training. These include overuse, insufficient rest periods after exercise, inadequate conditioning of the targeted body parts, excessive use of heavy loads, improper technique during certain exercises, and the abuse of performance-enhancing drugs [21]. We firmly feel and concur that in order to avoid injuries, it is crucial to provide close monitoring and guidance on proper technique and movement during this particular activity, ideally with the assistance of a trainer. Exercises that are more intricate may need trainers with more expertise, since the trainers' experience may significantly influence the trainees' musculoskeletal discomfort. According to research by Ahmed et al. (2022), fitness participants who received instruction from less experienced trainers had a risk of feeling discomfort at various locations that was more than double that of those who received instruction from more experienced trainers [22].

In Bangladesh, the general population has a musculoskeletal condition prevalence of approximately 30.4%, with low back pain at 18.6%, knee pain at 7.3%, and soft tissue

rheumatism at 3.8% being the top three conditions [27]. Similar to the findings, our study found that the most common musculoskeletal injuries were in the low back (17.2%), shoulder (11.0%), and knee (7.6%).

## Strength and limitations

Questionnaires given by face-to-face interviewers are more accurate and have more validity as they decrease non-response and misclassification bias. The results may be more broadly applicable as a result of the random sample technique used to include the whole population, which minimizes the selection bias. Incorporating the nationwide population of the country may enhance the generalizability of the results. However, there are also a number of limitations to this research. First off, this cross-sectional research cannot ascertain the cause of the correlation between the independent and dependent variables. Second, since this research was conducted in urban regions, its findings may not apply to Bangladesh's other rural communities. Furthermore, the questionnaire relied on some inquiries that necessitated participants to recollect past occurrences, thus resulting in recall bias. Future research should prioritize the collection of data pertaining to the characteristics of injuries and the specific treatment protocols necessary for their management.

## Conclusion

Gym-goers in Bangladesh often have muscular injuries. The back was the most often injured area, followed by the shoulder and knee. Individuals of the male gender and those aged 45 and above had elevated probabilities of experiencing musculoskeletal injury. In addition, those who visit the gym with the intention of weight loss and improving physical fitness are at a higher risk of experiencing musculoskeletal injuries. Individuals without sufficient understanding of workout knowledge, who did not exercise with caution, and did not get proper guidance from their trainer, were at greater risk of musculoskeletal injury. It is important to exhibit caution and take extra care while doing strength training activities in order to avoid injury. Prior to engaging in gym-based activities, it is essential to have a thorough understanding of proper exercise knowledge.

## Supporting information

**S1 Text. Questionnaire prevalence and predictors of musculoskeletal injuries among gym members in Bangladesh.**
(DOCX)

**S2 Text.**
(SAV)

**S1 Checklist. STROBE checklist.**
(DOCX)

## Acknowledgments

The authors are thankful to the participants for providing the information used to conduct the study.

## Author Contributions

**Conceptualization:** Mohammad Jahirul Islam, Sohel Ahmed.

**Data curation:** Mohammad Jahirul Islam, Md. Selim Rana, Md. Sharifuddin Sarker, Md. Mahemanul Islam, Md. Nuruzzaman Miah, Md. Anwar Hossain, Ruwaida Jahangir, Rahemun Akter, Sohel Ahmed.

**Formal analysis:** Md. Selim Rana, Sohel Ahmed.

**Funding acquisition:** Mohammad Jahirul Islam, Sohel Ahmed.

**Investigation:** Mohammad Jahirul Islam, Md. Selim Rana, Md. Sharifuddin Sarker, Md. Mahemanul Islam, Md. Nuruzzaman Miah, Md. Anwar Hossain, Ruwaida Jahangir, Rahemun Akter, Sohel Ahmed.

**Methodology:** Mohammad Jahirul Islam, Sohel Ahmed.

**Project administration:** Md. Selim Rana, Md. Sharifuddin Sarker, Md. Mahemanul Islam, Md. Nuruzzaman Miah, Md. Anwar Hossain, Ruwaida Jahangir, Sohel Ahmed.

**Resources:** Mohammad Jahirul Islam, Md. Selim Rana, Md. Sharifuddin Sarker, Md. Mahemanul Islam, Md. Nuruzzaman Miah, Md. Anwar Hossain, Ruwaida Jahangir, Sohel Ahmed.

**Software:** Sohel Ahmed.

**Supervision:** Sohel Ahmed.

**Validation:** Sohel Ahmed.

**Visualization:** Sohel Ahmed.

**Writing – original draft:** Rahemun Akter, Sohel Ahmed.

**Writing – review & editing:** Rahemun Akter, Sohel Ahmed.

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
