## [Decision Letter · Decision Letter 0]

28 Feb 2024

PONE-D-23-42987Prevalence and predictors of musculoskeletal injuries among gym members in Bangladesh: A nationwide cross-sectional study.PLOS ONE

Dear Dr. Ahmed,

Thank you for submitting your manuscript to PLOS ONE. After careful consideration, we feel that it has merit but does not fully meet PLOS ONE’s publication criteria as it currently stands. Therefore, we invite you to submit a revised version of the manuscript that addresses the points raised during the review process.

We look forward to receiving your revised manuscript.

Kind regards,

M Tanveer Hossain Parash, MBBS, M Phil

Academic Editor

PLOS ONE

Journal Requirements:

2. In the online submission form, you indicated that "Data will be made available prior to a request to the responding author."

5. Please include a copy of Table 2 which you refer to in your text on page 13.

Reviewers' comments:

Reviewer's Responses to Questions

**Comments to the Author**

1. Is the manuscript technically sound, and do the data support the conclusions?

Reviewer #1: Yes

Reviewer #2: No

2. Has the statistical analysis been performed appropriately and rigorously? 

Reviewer #1: Yes

Reviewer #2: No

3. Have the authors made all data underlying the findings in their manuscript fully available?

Reviewer #1: No

Reviewer #2: Yes

4. Is the manuscript presented in an intelligible fashion and written in standard English?

Reviewer #1: Yes

Reviewer #2: No

5. Review Comments to the Author

Reviewer #1: Reviewer Report:

This is an interesting study on musculoskeletal injuries among gym members in Bangladesh.

I propose to address the following minor issues.

Abstract:

It is better to report the number of participants and some statistical analytical procedure in the abstract.

Introduction:

Line 94: The author reported outcome from previous studies from Netherlands. However, the author needs to mention outcome from Bangladesh or similar countries (if available).

Line 104: The 2nd paragraph of introduction was written as discussion. In my opinion the 2nd paragraph needs to be rewritten.

Line 114: in third paragraph, the author again discusses the findings from Netherland and KSA. However, the needs to deduce proper rationale of the study and to focus on research gap in Bangladesh.

Methods:

Line 143: The author needs to mention the name and institution of the review board.

Line 172: The author needs to mention that the questionnaire that was used is validated in Bangladesh previously or not. If not, why they used unvalidated questionnaire.

The author also needs to provide details about inclusion and exclusion criteria.

Discussion:

Discussion is written proper order.

General Comments:

The language of writing need to be improved.

Reviewer #2: Method:

It can be assumed that there are many cities in Bangladesh. Why did the author choose only five cities? Could you please give the names of the cities and explain why these cities were selected over others? Additionally, why did the authors exclude gyms situated in semi-urban areas or villages? Is this selection of data sources convenient? If so, how can this study be considered a nationwide survey? Please provide a definition of "nationwide" in the method section to clarify why it was included in the title. Also, are "cities" and "divisions" synonymous in the context of Bangladesh?

Furthermore, why did the authors choose to withhold the name of the ethical review board? Is this board independent? Please provide details of the ethical approval process and consider including a copy of the approval as a supplementary file with the manuscript.

Result:

The results suggest that 22.9% of injured patients consulted with physiotherapists, a percentage surprisingly higher than that of physicians. However, this finding may not be realistic in the context of Bangladesh. Readers may question the validity of these results, necessitating a thorough examination of the data collection methods for potential biases.

Discussion:

What is the prevalence of musculoskeletal (MSK) issues in Bangladesh, and is there any evidence from previous studies to support this? No papers from Bangladesh have been cited in this study except for a self-citation. Did the authors search for MSK prevalence studies conducted in Bangladesh for reference? If such studies were not found, please mention this in the discussion section to provide context for the readers.

6. PLOS authors have the option to publish the peer review history of their article (what does this mean?). If published, this will include your full peer review and any attached files.

Reviewer #1: **Yes: **Md Kamruzzaman

Reviewer #2: No

---

## [Author Response · Author response to Decision Letter 0]

3 Mar 2024

Reply to the reviewer’s comments

We express our gratitude to the reviewer for their meticulous and in-depth reading of this manuscript, as well as for their insightful remarks and helpful recommendations, all of which have enabled us to enhance the text's quality.

Reviewer #1

Comment No: 1 It is better to report the number of participants and some statistical analytical procedure in the abstract.

Reply No: 1 We appreciate the reviewer's insightful feedback and the chance to improve the quality of our manuscript. We have improved our manuscript in the amended text in response to your feedback, and we have marked it with YELLOW.

Comment No: 2 Line 94: The author reported outcome from previous studies from Netherlands. However, the author needs to mention outcome from Bangladesh or similar countries (if available).

Reply No: 2 We appreciate your suggestion; however, the altered text now includes a reference to a similar Malaysian study. There isn't a comparable published work conducted in Bangladesh.

Comment No: 3 Line 104: The 2nd paragraph of introduction was written as discussion. In my opinion the 2nd paragraph needs to be rewritten.

Reply No: 3 Thank you for your valuable comment. Upgraded in the revised text.

Comment No: 4 Line 114: in third paragraph, the author again discusses the findings from Netherland and KSA. However, the needs to deduce proper rationale of the study and to focus on research gap in Bangladesh.

Reply No: 4 Thank you for your valuable comment. Upgraded in the revised text.

Comment No: 5 Line 143: The author needs to mention the name and institution of the review board.

Reply No: 5 The updated manuscript now includes the name of the ethical review board.

Comment No: 6 Line 172: The author needs to mention that the questionnaire that was used is validated in Bangladesh previously or not. If not, why they used unvalidated questionnaire.

Reply No: 6 A popular, trustworthy, and valid instrument for evaluating musculoskeletal injuries is the Nordic Musculoskeletal Disorder Questionnaire. There isn't a previous study that evaluates Bengali language cross-cultural adaptation and validation.

Comment No: 7 The author also needs to provide details about inclusion and exclusion criteria.

Reply No: 7 Men and women between the ages of 18 and 50 who have agreed to participate in the study and have been regular gym goers for at least six months are included in the sample. Participants who used steroids, had musculoskeletal or neurological disorders, or a history of non-gym-related injuries were excluded from the study.

Comment No: 8 Discussion is written proper order.

Reply No:8 Thank you for your positive feedback.

Reviewer #2:

Comment No: 1 It can be assumed that there are many cities in Bangladesh. Why did the author choose only five cities? Could you please give the names of the cities and explain why these cities were selected over others? Additionally, why did the authors exclude gyms situated in semi-urban areas or villages? Is this selection of data sources convenient? If so, how can this study be considered a nationwide survey? Please provide a definition of "nationwide" in the method section to clarify why it was included in the title. Also, are "cities" and "divisions" synonymous in the context of Bangladesh?

Reply No: 1 We appreciate the reviewer's insightful feedback and the chance to improve the quality of our manuscript. We have improved our manuscript in the amended text in response to your feedback, and we have marked it with GREEN.

Data was collected from 74 fitness facilities from various divisional cities, districts, and gym centers at the thana level in five divisions of Bangladesh (Dhaka, Sylhet, Rajshahi, Khulna, and Chittagong). We made a random visit to gym facilities because the number of gym centers in Bangladesh and the number of people who regularly frequent gyms are not well-documented. The fact that this study was carried out in urban areas limits its applicability to other rural groups in Bangladesh. Potential samples were included using a dual-stage cluster sampling technique and a nationwide sample consisting of five administrative divisions of Bangladesh. The authors express their gratitude to the reviewer for bringing to our attention our inadvertent error with the word choice "cities." To be clear, we gathered information from 74 fitness centers spread over five administrative divisions, including metropolitan and semi-urban areas.

Comment No: 2 Furthermore, why did the authors choose to withhold the name of the ethical review board? Is this board independent? Please provide details of the ethical approval process and consider including a copy of the approval as a supplementary file with the manuscript.

Reply No: 2 Blinded peer review required many journals; therefore, we prepared it appropriately. We also submitted the ethical approval letter as an additional file with the submission, and we have now included the details in the updated text.

Comment No: 3 The results suggest that 22.9% of injured patients consulted with physiotherapists, a percentage surprisingly higher than that of physicians. However, this finding may not be realistic in the context of Bangladesh. Readers may question the validity of these results, necessitating a thorough examination of the data collection methods for potential biases.

Reply No: 3 We concur with the reviewer's comment. The participants in our study were individuals with a keen interest in physical activity and sports. Today, those interested in sports and physical activities are more acquainted with physiotherapy, possibly explaining the trend.

Comment No: 4 What is the prevalence of musculoskeletal (MSK) issues in Bangladesh, and is there any evidence from previous studies to support this? No papers from Bangladesh have been cited in this study except for a self-citation. Did the authors search for MSK prevalence studies conducted in Bangladesh for reference? If such studies were not found, please mention this in the discussion section to provide context for the readers.

Reply No: 4 Thank you for addressing this significant matter. Our study focuses on musculoskeletal injuries among gym members and may not be applicable to the entire population in Bangladesh. We have included a recent article on musculoskeletal issues among Bangladeshi people. There is a lack of literature on musculoskeletal injuries among the general population or gym members in Bangladesh.

---

## [Decision Letter · Decision Letter 1]

3 Apr 2024

PONE-D-23-42987R1Prevalence and predictors of musculoskeletal injuries among gym members in Bangladesh: A nationwide cross-sectional study.PLOS ONE

Dear Dr. Ahmed,

Thank you for submitting your manuscript to PLOS ONE. After careful consideration, we feel that it has merit but does not fully meet PLOS ONE’s publication criteria as it currently stands. Therefore, we invite you to submit a revised version of the manuscript that addresses the points raised during the review process.

We look forward to receiving your revised manuscript.

Kind regards,

M Tanveer Hossain Parash, MBBS, M Phil

Academic Editor

PLOS ONE

Additional Editor Comments:

1. Please incorporate the following comments you sent in email "Data were collected from all five geographic divisions of Bangladesh, and a dual-stage cluster sampling technique was used to include potential samples. We randomly chose the gym center as clusters in the first stage. Due to the lack of available data on the total number of gym centers in Bangladesh and the number of individuals who regularly attend gyms, we conducted a random visit to gym facilities in five divisional cities. In order to streamline data collection, we made a single visit to each gym facility and gathered data from the fitness trainees who were there at the time." under the sampling sub-section.

2. Please provide description in the result section how your data met the following assumptions of the logistic regression "Basic assumptions that must be met for logistic regression include independence of errors, linearity in the logit for continuous variables, absence of multicollinearity, and lack of strongly influential outliers."

Reviewers' comments:

Reviewer's Responses to Questions

**Comments to the Author**

1. If the authors have adequately addressed your comments raised in a previous round of review and you feel that this manuscript is now acceptable for publication, you may indicate that here to bypass the “Comments to the Author” section, enter your conflict of interest statement in the “Confidential to Editor” section, and submit your "Accept" recommendation.

Reviewer #1: All comments have been addressed

Reviewer #3: (No Response)

2. Is the manuscript technically sound, and do the data support the conclusions?

Reviewer #1: Yes

Reviewer #3: Partly

3. Has the statistical analysis been performed appropriately and rigorously? 

Reviewer #1: Yes

Reviewer #3: Yes

4. Have the authors made all data underlying the findings in their manuscript fully available?

Reviewer #1: Yes

Reviewer #3: No

5. Is the manuscript presented in an intelligible fashion and written in standard English?

Reviewer #1: Yes

Reviewer #3: Yes

6. Review Comments to the Author

Reviewer #1: The manuscript has now been corrected and improved properly. All comments have been addressed properly

Reviewer #3: The current state of the manuscript provides fewer significant findings or new knowledge on the prevalence and predictors of musculoskeletal injuries among gym members in Bangladesh. The author should focus on highlighting which parts of the findings are relevant to add to the current knowledge. The discussion and conclusion provide very general and well-known information. Please find the comments on the attached draft.

7. PLOS authors have the option to publish the peer review history of their article (what does this mean?). If published, this will include your full peer review and any attached files.

Reviewer #1: **Yes: **Md Kamruzzaman

Reviewer #3: No

---

## [Author Response · Author response to Decision Letter 1]

4 Apr 2024

Manuscript Title: Prevalence and predictors of musculoskeletal injuries among gym members in Bangladesh: A nationwide cross-sectional study.

Reply to the reviewer’s comments

We express our gratitude to the reviewer for their meticulous and in-depth reading of this manuscript, as well as for their insightful remarks and helpful recommendations, all of which have enabled us to enhance the text's quality.

Additional Editor Comments

Comment No: 1 Please incorporate the following comments you sent in email "Data were collected from all five geographic divisions of Bangladesh, and a dual-stage cluster sampling technique was used to include potential samples. We randomly chose the gym center as clusters in the first stage. Due to the lack of available data on the total number of gym centers in Bangladesh and the number of individuals who regularly attend gyms, we conducted a random visit to gym facilities in five divisional cities. In order to streamline data collection, we made a single visit to each gym facility and gathered data from the fitness trainees who were there at the time." under the sampling sub-section.

Reply No: 1 We appreciate the insightful feedback and the chance to improve the quality of our manuscript. The information has been added in the revised text.

Comment No: 2 Please provide description in the result section how your data met the following assumptions of the logistic regression "Basic assumptions that must be met for logistic regression include independence of errors, linearity in the logit for continuous variables, absence of multicollinearity, and lack of strongly influential outliers."

Reply No: 2 

In the results section, we found that our data met the basic assumptions required for logistic regression analysis:

Independence of Errors: We evaluated the independence of error by analyzing the residuals (the discrepancies between observed and anticipated values). Our analysis revealed no significant patterns or correlations, suggesting that the assumption of error independence was satisfied.

Linearity in the Logit for Continuous Variables: We analyzed the correlation between the predictor factors and the log odds of the outcome using scatterplots and partial residual plots. The plots demonstrated a direct correlation between the continuous predictor variables and the log odds of the outcome This suggests that the assumption of linearity in the logit model was satisfied.

Absence of Multicollinearity: We assessed multicollinearity among the predictor variables using variance inflation factor (VIF) values. VIF values below 5.0 indicated that multicollinearity was not a concern, suggesting that the assumption of absence of multicollinearity was met.

Lack of Strongly Influential Outliers: We analyzed important outliers by using diagnostic plots such as Cook's distance and leverage plots. The plots demonstrated that there were no observations that had an excessively significant impact on the regression coefficients, confirming that the premise of the absence of very influential outliers was satisfied.

Overall, our data met the assumptions of independence of errors, linearity in the logit for continuous variables, absence of multicollinearity, and lack of strongly influential outliers, supporting the validity of our logistic regression analysis.

The following statement has been added in the revised text “The presence of multicollinearity among the independent variables was assessed using variance inflation factors (VIFs), with a predetermined threshold of VIF≤5.0.’’

Reviewer #3

We express our gratitude to the reviewer for their meticulous and comprehensive evaluation of this paper, as well as for their insightful remarks and valuable recommendations, which significantly contributed to enhancing the overall quality of this work.

Comment No: 1 This statement contradicts the results presented in Table 1. According to Table 1, participants who went to the gym for physical fitness are less likely to experience pain, while those who went to lose weight experience pain. Does this statement refer to Table 2? How did the author correlate data in Table 1 and Table 2? I find it challenging to understand the significant information or new findings that the author intended to highlight in this study. The author should be clearer and provide more elaboration on these findings. 

I suggest the author to put the Table number or Figure number on each statement in the results and discussion section to make it easier for the reader to understand.

Reply No: 1 Thank you for your insightful comment. The statement has been modified in the revised text.

Comment No: 2 Please put reference number for this statement.

Reply No: 2 Reference has been added in the revised text

Comment No: 3 I suggest the author to rewrite this paragraph. The sample size for males versus females in the current study is not comparable since the male population (n=1039) is much higher than the female population (n=90). Therefore, discussing the significance of injuries among men versus women is irrelevant.

Reply No: 3 We agree with the reviewer's comment that our study had a majority of male participants. However, we have discussed the statistical findings of our study without regard to the number of participants.

Comment No: 4 This statement should be elaborated further on the discussion.

Reply No: 4 We appreciate your attention to this matter and your contribution to enhancing the quality of our text. The issue has been addressed in the first paragraph of the updated text.

Comment No: 5 This sentence is a common knowledge and I don't think this is a significant finding that should be highlighted on the conclusion.

Reply No: 5 Thank you for your structured comment. The following statement has been added in the revised text “Individuals without sufficient understanding of workout knowledge, who did not exercise with caution, and did not get proper guidance from their trainer, were at greater risk of musculoskeletal injury’’.

---

## [Decision Letter · Decision Letter 2]

25 Apr 2024

Prevalence and predictors of musculoskeletal injuries among gym members in Bangladesh: A nationwide cross-sectional study.

PONE-D-23-42987R2

Dear Dr. Ahmed,

We’re pleased to inform you that your manuscript has been judged scientifically suitable for publication and will be formally accepted for publication once it meets all outstanding technical requirements.

Kind regards,

M Tanveer Hossain Parash, MBBS, M Phil

Academic Editor

PLOS ONE

Additional Editor Comments (optional):

Reviewers' comments:

Reviewer's Responses to Questions

**Comments to the Author**

1. If the authors have adequately addressed your comments raised in a previous round of review and you feel that this manuscript is now acceptable for publication, you may indicate that here to bypass the “Comments to the Author” section, enter your conflict of interest statement in the “Confidential to Editor” section, and submit your "Accept" recommendation.

Reviewer #3: All comments have been addressed

2. Is the manuscript technically sound, and do the data support the conclusions?

Reviewer #3: Yes

3. Has the statistical analysis been performed appropriately and rigorously? 

Reviewer #3: Yes

4. Have the authors made all data underlying the findings in their manuscript fully available?

Reviewer #3: Yes

5. Is the manuscript presented in an intelligible fashion and written in standard English?

Reviewer #3: Yes

6. Review Comments to the Author

Reviewer #3: The current state of the manuscript provides enough information for publication. All comments from the previous version have been addressed by the author.

7. PLOS authors have the option to publish the peer review history of their article (what does this mean?). If published, this will include your full peer review and any attached files.

Reviewer #3: No

---

## [Editor Report · Acceptance letter]

24 Jul 2024

PONE-D-23-42987R2 

PLOS ONE

Dear Dr. Ahmed, 

I'm pleased to inform you that your manuscript has been deemed suitable for publication in PLOS ONE. Congratulations! Your manuscript is now being handed over to our production team.

Kind regards, 

on behalf of

Dr. M Tanveer Hossain Parash 

Academic Editor

PLOS ONE